# Serum Neurofilament Light Chain as Biomarker for Cladribine-Treated Multiple Sclerosis Patients in a Real-World Setting

**DOI:** 10.3390/ijms24044067

**Published:** 2023-02-17

**Authors:** Michael Seiberl, Julia Feige, Patrick Hilpold, Wolfgang Hitzl, Lukas Machegger, Arabella Buchmann, Michael Khalil, Eugen Trinka, Andrea Harrer, Peter Wipfler, Tobias Moser

**Affiliations:** 1Department of Neurology, Christian Doppler University Hospital, Paracelsus Medical University and Center for Cognitive Neuroscience, European Reference Network EpiCARE, 5020 Salzburg, Austria; 2Research Management (RM): Biostatistics and Publication of Clinical Studies Team, Paracelsus Medical University, 5020 Salzburg, Austria; 3Department of Ophthalmology and Optometry, Paracelsus Medical University, 5020 Salzburg, Austria; 4Research Program Experimental Ophthalmology and Glaucoma Research, Paracelsus Medical University, 5020 Salzburg, Austria; 5Division of Neuroradiology, Christian Doppler Medical Center, Paracelsus Medical University, 5020 Salzburg, Austria; 6Department of Neurology, Medical University of Graz, 8036 Graz, Austria; 7Neuroscience Institute, Christian Doppler University Hospital, Paracelsus Medical University and Center for Cognitive Neuroscience, 5020 Salzburg, Austria; 8Department of Dermatology and Allergology, Paracelsus Medical University, 5020 Salzburg, Austria

**Keywords:** disease activity, NfL, treatment response, biomarkers, multiple sclerosis, immune reconstitution

## Abstract

Serum neurofilament light chain (sNfL) is an intensely investigated biomarker in multiple sclerosis (MS). The aim of this study was to explore the impact of cladribine (CLAD) on sNfL and the potential of sNfL as a predictor of long-term treatment response. Data were gathered from a prospective, real-world CLAD cohort. We measured sNfL at baseline (BL-sNfL) and 12 months (12Mo-sNfL) after CLAD start by SIMOA. Clinical and radiological assessments determined fulfilment of “no evidence of disease activity” (NEDA-3). We evaluated BL-sNfL, 12M-sNfL and BL/12M sNfL ratio (sNfL-ratio) as predictors for treatment response. We followed 14 patients for a median of 41.5 months (range 24.0–50.0). NEDA-3 was fulfilled by 71%, 57% and 36% for a period of 12, 24 and 36 months, respectively. We observed clinical relapses in four (29%), MRI activity in six (43%) and EDSS progression in five (36%) patients. CLAD significantly reduced sNfL (BL-sNfL: mean 24.7 pg/mL (SD ± 23.8); 12Mo-sNfL: mean 8.8 pg/mL (SD ± 6.2); *p* = 0.0008). We found no correlation between BL-sNfL, 12Mo-sNfL and ratio-sNfL and the time until loss of NEDA-3, the occurrence of relapses, MRI activity, EDSS progression, treatment switch or sustained NEDA-3. We corroborate that CLAD decreases neuroaxonal damage in MS patients as determined by sNfL. However, sNfL at baseline and at 12 months failed to predict clinical and radiological treatment response in our real-world cohort. Long-term sNfL assessments in larger studies are essential to explore the predictive utility of sNfL in patients treated with immune reconstitution therapies.

## 1. Introduction

Cladribine (CLAD) acts as immune reconstitution therapy (IRT) and is used for the treatment of multiple sclerosis (MS) [1,2]. IRTs induce a peripheral lymphodepletion with the aim of correcting the immunological dysregulations in MS [2]. CLAD tablets are given as short cycles in year one and two followed by a drug-free interval of up to several years [3]. The timing of treatment re-initiation, however, needs to be explored further. While patients with stable disease should not be exposed to potential adverse effects associated with additional immunomodulation, long-term sequalae in case of disease recurrence, on the other hand, should also be avoided. To achieve this goal, robust indicators to predict individual treatment response are needed.

Serum neurofilament light chain protein (sNfL) is an easily accessible serum biomarker of neuroaxonal damage [4]. SNfL correlates with clinical and radiological disease activity in MS, and baseline values hold the potential to predict disability accumulation in patients with clinically isolated syndrome [5,6,7]. These data support a role of sNfL as both a readout for treatment response and also a tool to capture long-term disease outcome.

Understanding the efficacy and durability of pulsed immune reconstitution by CLAD is instrumental in guiding an optimal treatment concept for the individual MS patient. We therefore aimed to study the impact of oral CLAD on sNfL and its potential as a predictor of long-term treatment response among our real-world cohort.

## 2. Results

We included 14 patients with relapsing MS, with a mean age of 35.4 years (± 9.3) and a median EDSS of 1.8 (interquartile range (IQR) 1.0–2.4, range 0–3.5) at BL. Two patients were treatment-naïve, and nine patients had experienced a clinical relapse in the three months before CLAD start. Patients were followed-up for a median of 41.5 months (IQR 36.0–46.3, range 24.0–50.0) after CLAD initiation. Demographic features and cohort characteristics are displayed in Table 1, while Appendix A summarizes the disease course of each patient.

NEDA-3 was fulfilled by 71% (10/14), 57% (8/14) and 36% (4/11) of patients for a period of 12, 24 and 36 months, respectively (Figure 1). Five (36%) patients had ongoing high disease activity, whilst the majority (64%) showed no or only mild signs of disease activity throughout the follow-up (Appendix A). Overall, 10 relapses were recorded in 4 patients (29%) whereas 10 patients (71%) remained relapse-free. In addition to cerebral MRI, radiological investigations included the cervical spine in 9/14 patients (64%) and the thoracic spine in one patient.

MRI activity was observed in 6/14 (43%). Throughout the follow-up, EDSS remained stable in five (36%), improved in four (29%) and worsened in five (36%) patients. Median EDSS improved from 1.8 (interquartile range (IQR) 1.0–2.4, range 0–3.5) at BL to 1.3 (interquartile range (IQR) 0.0–2.4, range 0–6.5) at study termination. Due to ongoing disease activity, 4/14 patients were switched to other immunotherapies (Table 1), while no patient received more than two CLAD cycles.

Serum NfL–impact of CLAD and predictive value: Mean sNfL significantly decreased under CLAD therapy from 24.7 pg/mL (±23.8) at baseline to 8.8 pg/mL (±6.2) at 12 months (*p* = 0.0008; Figure 2a). This mean reduction of 15.9 pg/mL (95% CI: 5.5–25.5) amounts to an overall decrease in sNfL of 65%. SNfL dropped in 13/14 (93%) patients after CLAD initiation and was stable in the remaining individual.

Additionally, we investigated whether short-term sNfL assessment could be used to predict long-term disease control on an individual basis. We found no correlation between BL-sNfL (*p* = 0.88), 12Mo-sNfL (*p* = 0.36), ratio-sNfL (*p* = 0.91) and time-to-loss of NEDA-3. Moreover, there was no correlation between sNfL parameters and occurrence, number and severity of relapses. Also, MRI activity, sustained disability worsening, therapy switch and fulfilment of NEDA-3 did not correlate with sNfL (Figure 2b–f). Finally, we found no correlation of either sNfL parameter and whether patients were treatment-naïve at the time of or had a disease-modifying therapy (DMT) before cladribine start. The type of pre-treatment (first or second line) had no impact on the sNfL course.

## 3. Discussion

In the present study, while exploring the effect of oral CLAD on sNfL, we found a marked reduction of sNfL, emphasizing the beneficial role of IRTs on brain damage and the potential of CLAD to attenuate MS disease activity. Mean sNfL levels dropped from 25 pg/mL to below 10 pg/mL after the first CLAD treatment year, resulting in decreases to values back to levels of healthy controls [6,8]. SNfL reductions among our cohort were clinically accompanied by amelioration of median EDSS. Our results are in line with a recent Italian study, which observed a similar strong reduction of sNfL at 24 weeks among 18 MS patients [9]. While the exact mode of action of CLAD remains elusive, the impact on central nervous system inflammation appears mainly attributed to the peripheral consequences of the drug on circulating lymphocytes [10]. Th17 subsets and memory B cells, both major culprits in MS pathogenesis [11,12], are reduced by CLAD, followed by an incomplete recovery at 24 months [13]. CLAD also affects inflammatory cell adhesion molecules expressed by leucocytes, which may influence lymphocyte communication and migration [14]. Moreover, CLAD itself can enter the central nervous system where it may attenuate local inflammation [15]. In fact, subcutaneous administration of CLAD was shown to eliminate oligoclonal bands in the cerebrospinal fluid (CSF) of people with MS (pwMS) [16]. 

Reductions in sNfL have been attributed to several DMTs for continuous administration and are increasingly considered as outcome measures in drug trials [17,18,19]. A study on patients treated with alemtuzumab, another IRT, found that sNfL levels not only decreased close to physiological levels, but also showed a sustained effect for up to seven years [20]. Together with clinical and MRI outcome parameters, these data underline the long-term potential of IRTs to attenuate pathological processes. Congruent reductions in sNfL were also reported in MS patients receiving autologous hematopoietic stem cell therapy, which is considered the strongest IRT [21]. 

While there was a clear impact of CLAD on sNfL, the values at BL and at 12 months were not predictive of the long-term disease outcome among this explorative cohort. Also, the relative reduction of sNfL associated with the first cycle of CLAD did not correlate with clinical and radiological treatment response. To summarize, sNfL assessment within the first year of treatment failed to predict sustained disease control among this real-world CLAD cohort. In line with our data, sNfL was not able to capture or predict EDSS progression that occurred independent of relapse or MRI activity in natalizumab-treated patients [22]. On the other hand, lower BL-sNfL correlated with a better treatment response in patients treated with dimethyl fumarate, a first-line MS drug [23]. NfL has been investigated in several neurological conditions and was found to be more sensitive to acute axonal loss than to sustained damage that underlies neurodegenerative processes [24]. This could explain why sNfL appears especially useful to predict disease outcome when assessed during the initial phase of the disease which is characterized by acute focal inflammation [6]. In fact, natalizumab and CLAD represent second-line therapies often prescribed after failure of other DMTs. It can be hypothesised that, by the time these treatment options are initiated, neurodegenerative processes may be more prominent [25]. Therefore, the predictive value of sNfL in later stages of MS has to be explored further.

During the mean follow-up of more than three years, the majority of our patients had no relapse and no confirmed disability progression, supporting the hypothesis that short-term CLAD administration induces a sustained immunological reset. In line with our data, the durable efficacy of CLAD has been confirmed in pivotal trials and corroborated by real-world data [1,26,27,28], underpinning the role of CLAD as an effective and convenient treatment option for patients with MS.

The SARS-CoV-2 pandemic prohibited assessment of sNfL samples beyond the first year which represents a major limitation of this work. Studies assessing the course of sNfL over a longer period are currently being conducted and will provide further insights into the sustainability of CLAD-induced treatment effects [29]. Whether evaluation of the sNfL course at regular intervals can be a useful tool to detect subclinical disease recurrence, as suggested in a pilot study among patients receiving alemtuzumab [30], has to be investigated further. Future studies will show whether long-term sNfL assessment may have an important role in determining the optimal time point for each individual patient to re-initiate treatment following IRT. Another open question is whether repopulation kinetics of specific lymphocyte subsets correlate with disease activity, and in-depth immunophenotyping analyses beyond years 1 and 2 are yet to be performed. Aside from the merely short-term assessment of sNfL, the size of our cohort represents another major limitation. Moreover, MRI analyses of the spine were only available in a minority of patients, and we can therefore not exclude clinically silent inflammations in the spinal cord with impacts on sNfL.

## 4. Materials and Methods

Patients were enrolled from a prospective CLAD-treated real-world study initiated in 2017 and conducted at the outpatient MS clinic of the Medical University Salzburg. This patient cohort had been primarily recruited to explore the effect of CLAD on pathogen-specific antibody levels and consisted of fourteen patients with relapsing MS. We collected demographics and patient history at the time of CLAD start (baseline, BL) and clinical as well as magnetic resonance imaging (MRI) data. Patient were re-evaluated by MS specialists every 3–6 months within the first two treatment years and every 3–12 months thereafter. We assessed clinical relapses as well as sustained disability worsening defined as 6 month confirmed Expanded Disability Status Scale (EDSS) progression. Severe relapses were defined as motor or brainstem symptoms and steroid-refractory optic neuritis. We recorded immunotherapy switches following CLAD treatment. Cerebral MRI was performed at least annually on 3 tesla MRI devices and included T1-weighted images before and after administration of contrast agent (gadolinium—Gd) and T2/fluid-attenuated inversion recovery (FLAIR) sequences. Spine MRI evaluations were also considered if available. Images were analysed by two independent neuroradiologists. MRI activity was defined as new or enlarged T2/FLAIR lesions, or T1 gadolinium enhancement. We considered the NEDA-3 status consisting of the absence of clinical relapse, MRI activity, and sustained disability worsening [31].

SNfL assessment: Venous blood was collected at BL (BL-sNfL) and after 12 months (before initiation of the second CLAD cycle, 12Mo-sNfL), centrifuged for 10 minutes with 3000× *g* at room temperature, and serum aliquots were stored at −80 °C before shipment. SNfL was measured by single-molecule array (SIMOA, Quanterix Corporation, Lexington, MA, USA) assay NF-light^®^ advantage kit on the SR-X Analyzer (Quanterix Corporation, Lexington, MA, USA) at the Department of Neurology of the Medical University Graz. All samples were analysed as one single batch in order to improve data quality and comparability by personnel blinded for the clinical and MRI data.

Treatment response: The impact of CLAD on sNfL concentrations was calculated using the mean absolute sNfL values at baseline (BL-sNfL) and at 12 months from CLAD start (12Mo-sNfL). The utility of sNfL as a predictor for long-term treatment response was explored using BL-sNfL and 12Mo-sNfL concentrations and the ratio between both timepoints (ratio-sNfL, calculated with the formula (12Mo-sNfL–BL-sNfL)/(BL-sNfL*100), which expresses the relative change of sNfL associated with CLAD administration. We performed a time-to-event analysis in order to investigate whether loss of “no evidence of disease activity” (NEDA) correlated with either one of the three sNfL parameters. We also explored whether BL-sNfL, 12Mo-sNfL and/or ratio-sNfL were predictive for occurrence of relapses, MRI activity, EDSS progression, therapy switch or for sustained NEDA-3 throughout follow-up.

Statistics: Data were analysed for consistency, normality, and variance homogeneity. Due to small sample size, independent and dependent bootstrap t tests were computed. Kaplan–Meier analyses were used for time-to-event analyses and Spearman’s correlations for correlations. Whisker plots with 95% confidence intervals for means illustrate results. All reported tests were two-sided, and *p*-values < 0.05 were considered statistically significant. Statistical analyses in this report were performed by use of NCSS (NCSS 10, NCSS, LLC. Kaysville, UT), STATISTICA 13 (Hill, T. & Lewicki, P. Statistics: Methods and Applications. StatSoft, Tulsa, OK). Data are presented as mean (±standard deviation (SD)) unless otherwise stated.

Ethics: The study was approved by the local ethics committee (Ethics Committee of Salzburg 415-E/1612/11-2018) and conducted according to Good Clinical Practice and the ethical principles of the Declaration of Helsinki. Informed consent was obtained from all subjects involved in the study.

## 5. Conclusions

CLAD reduces neuroaxonal damage in patients with MS as reflected by sNfL and induces clinical disease control in many patients. Among our real-world cohort, sNfL at BL and after 12 months fails to predict sustained clinical and radiological disease control. Long-term sNfL evaluations should be assessed in multicentre studies to explore the potential of sNfL to predict treatment response and to capture the necessity of resuming therapy after immune reconstitution.

## Figures and Tables

**Figure 1 ijms-24-04067-f001:**
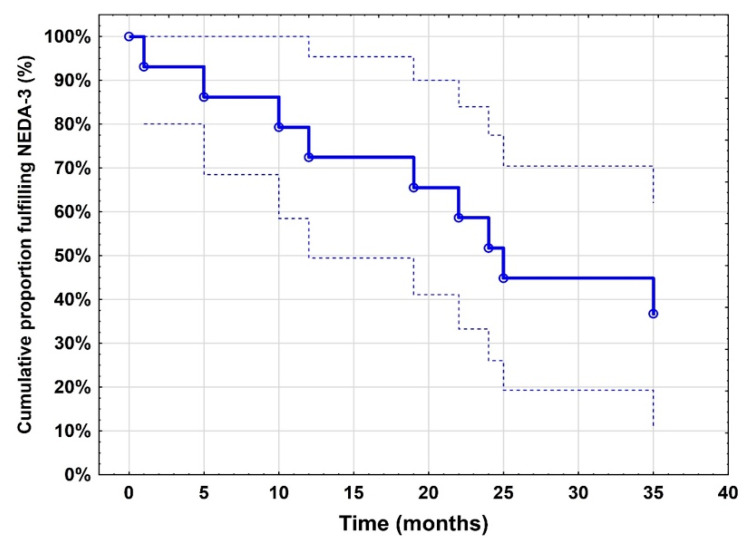
Kaplan–Meier curve showing sustained NEDA-3 fulfilment among the study cohort (*n* = 14). NEDA: no evidence of disease activity.

**Figure 2 ijms-24-04067-f002:**
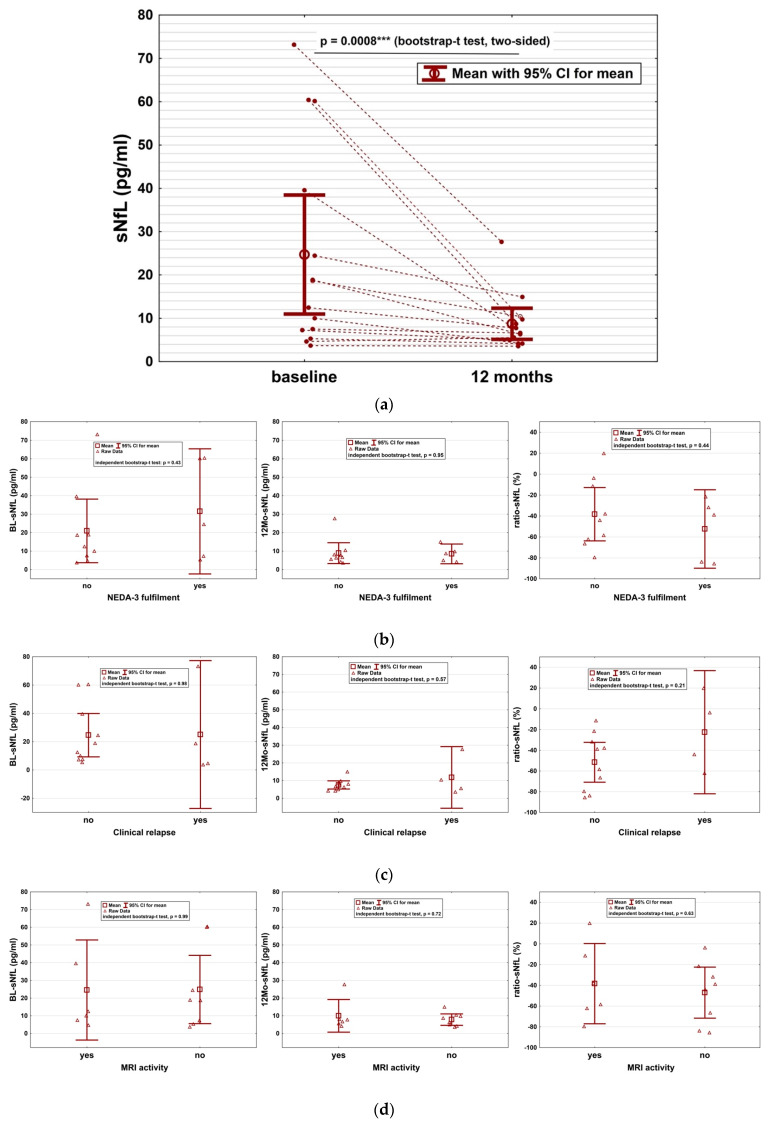
Impact of CLAD on sNfL and predictive utility of sNfL for treatment response. (**a**) SNfL is significantly reduced by CLAD administration (*p* = 0.0008) at 12 months from therapy start. (**b**–**f**) We found no correlation between sNfL at BL (BL-sNfL, left), sNfL at 12 months from CLAD start (12Mo-sNfL, middle) and the sNfL ratio of both time points (ratio-sNfL, right) and various parameters for disease activity. EDSS progression was defined as 6 months confirmed Expanded Disability Status Scale (EDSS) progression. CI: confidence interval; NEDA: no evidence of disease activity; MRI: magnetic resonance imaging; EDSS: Expanded Disability Status Scale; DMT: disease modifying therapy.

**Table 1 ijms-24-04067-t001:** Characteristics of patients included (*n* = 14).

Age (y), mean (SD)	35.4 (9.3)
Female, No. (%)	12 (86)
No. RRMS at baseline (%)	14 (100)
Median follow-up in months (IQR)	41.5 (36.0–46.3)
Median EDSS (range) at BL	1.8 (0–3.5)
Median EDSS (range) at EOS	1.3 (0–6.5)
Mean disease duration at BL, y (SD)	7.9 (7.4)
Patients with DMTs before CLAD (%) First-line DMTs (%) Second-line DMTs (%) No. DMTs before CLAD, median (min; max)	12/14 (86) 9/12 (75) 3/12 (25) 1.8 (0; 4)
Patients with DMTs after CLAD (%) Anti-CD20 infusion (%) S1P-modulator (%) Months from CLAD to DMT switch, mean (SD)	4/14 (29) 3 (75) 1 (25) 32 (13)

Y: years; SD: standard deviation; No.: number; RRMS: relapsing remitting multiple sclerosis; IQR: interquartile range; EDSS: Expanded Disability Status Scale; BL: baseline; EOS: end of study; CLAD: cladribine; DMT: disease modifying therapy.

## Data Availability

The data that support the findings of this study are available from the corresponding author (TM), upon reasonable request.

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
