# Peer review of "Serum Neurofilament Light Chain as Biomarker for Cladribine-Treated Multiple Sclerosis Patients in a Real-World Setting"

_ijms, 2023, doi:10.3390/ijms24044067_

Round 1

Reviewer 1 Report

This study investigated the role of sNfL as a biomarker for cladribine-treated multiple sclerosis patients in a real-world setting. Overall, the manuscript is well-written. I have several comments. 

1. As the authors indicated in the manuscript, the absence of a spine MRI can be a significant limitation of the study. The number of patients who were taken spine MRI should be clearly addressed.

2. The authors described that SNfL dropped in 13/14 (93%) patients. The criteria for sNfL decrease should be clarified. (% from baseline).

Author Response

  1. As the authors indicated in the manuscript, the absence of a spine MRI can be a significant limitation of the study. The number of patients who were taken spine MRI should be clearly addressed.
  2. The authors described that SNfL dropped in 13/14 (93%) patients. The criteria for sNfL decrease should be clarified. (% from baseline).

We really want to thank the reviewer for the time and for the comments.

Ad 1) Thank you, we agree that this information is of particular interest for the reader and have added the following paragraph to the results section:

Lines: 72-74

"In addition to cerebral MRI, radiological investigations included the cervical spine in 9/14 patients (64%) and the thoracic spine in one patient."

Ad 2) Thank you, this is of course a very important point.

We have now added the criteria for the decrease in the methods section and clarified the results section. The respective paragraphs now read as follows:

Methods:

Lines: 215-217

"The impact of CLAD on sNfL concentrations was calculated using the mean absolute sNfL values at baseline (BL-sNfL) and at 12 months from CLAD start (12Mo-sNfL)."

Results:

Lines: 86-90

"Mean sNfL significantly decreased under CLAD therapy from 24.7 pg/ml (±23.8) at baseline to 8.8 pg/ml (±6.2) at 12 months (p = 0.0008; figure 2a). This mean reduction of 15.9 pg/ml (95% CI: 5.5-25.5) amounts to an overall decrease in sNfL of 65%. SNfL dropped in 13/14 (93%) patients after CLAD initiation and was stable in the remaining individual."

Reviewer 2 Report

The authors reported herein the role of cladribine treatment of multiple sclerosis patients and its impact on the accumulation of neurofilament light chain protein in the blood. The authors nicely and comprehensibly addressed this issue in the provided manuscript. However, several points should deserve further attention of the authors as detailed below:

-       It is not clear from the data how much the treatment with first/second line DMTs (which some of the patients were having) affected the accumulation of NFL in the blood. Does cladribrine provide any additional/synergistic benefit in these patients? In other words, where the naïve patients medicated with first/second line DMTs?

-       Can the authors predict (or statistically correlate), the levels of NFL in the blood with the number/severity of relapses?

Author Response

We really want to thank the reviewer for the time and for the comments.

1) It is not clear from the data how much the treatment with first/second line DMTs (which some of the patients were having) affected the accumulation of NFL in the blood. Does cladribrine provide any additional/synergistic benefit in these patients? In other words, where the naïve patients medicated with first/second line DMTs?

Thank you, this is an interesting point. We found no impact on DMTs before CLAD and sNfL and have added the following paragraph to the results section (Lines: 96-99):

"Finally, we found no correlation of either sNfL parameter and whether patients were treatment naïve at the time of or had a disease modifying therapy (DMT) before cladribine start. The type of pre-treatment (first or second line) had no impact on the sNfL course."

2) Can the authors predict (or statistically correlate), the levels of NFL in the blood with the number/severity of relapses?

Thank you. Blood NfL was not able to predict number and severity of relapses among our cohort. Fort better clarification, we have now changed/Added as follows:

We have added the following definition to the methods section (Lines 198-199):

"Severe relapses were defined as motor or brainstem symptoms and steroid-refractory optic neuritis."

We have rephrased the respective paragraph which now reads as follows (Lines 94-96):

"Moreover, there was no correlation between sNfL parameters and occurrence, number and severity of relapses. Also, MRI activity, sustained disability worsening, therapy switch and fulfilment of NEDA-3 did not correlate with sNfL (figures 2b-f)."